# Horticultural Practices in Early Spring to Mitigate the Adverse Effect of Low Temperature on Fruit Set in ‘Lapins’ Sweet Cherry

**DOI:** 10.3390/plants12030468

**Published:** 2023-01-19

**Authors:** Hao Xu, Danielle Ediger, Mehdi Sharifi

**Affiliations:** Summerland Research and Development Centre, Agriculture and Agri-Food Canada, Summerland, BC V0H 1Z0, Canada

**Keywords:** anthesis, cold injury, fruit set, ovule abortion, pistil browning, pollen germination, soil temperature, warming

## Abstract

Yield of sweet cherry (*Prunus avium* L.) is determined by fruit set, a developmental stage sensitive to variable spring environmental conditions. To sustain fruit production and enhance crop climate resilience, it is important to understand the impacts of abiotic stresses and the effectiveness of horticultural mitigations in the spring on the critical developmental processes during fruit set. In this study, flowering phenology, pistil browning and percent fruit set of ‘Lapins’ were monitored at five sites of different elevation and frost risk in the Okanagan Valley, British Columbia, Canada, in 2019 and 2022. At Site 1 in Summerland Research and Development Centre (“SuRDC1”), where a ‘Lapins’ on Krymsk 5 planting was located in a frost pocket where the crops were exposed to high risk of cold damage in the spring, a series of experiments were conducted to investigate the floral organ viability and percent fruit set under low temperatures, and under the effects of four spring horticultural mitigation measures. Installation of polyethylene sleeves and FAME spray (fatty acid methyl esters-based plant growth regulator, WAIKEN, SST Australia) were implemented in 2019; boric acid spray and postponed irrigation were tested in 2022. Low fruit set at SuRDC1 in both years was associated with severe pistil browning after night temperature dropped below −4 °C in late April. In 2019, the semi-enclosure of polyethylene sleeves led to an increase in the surface temperature (T_surfae_) of floral buds by 2–4 °C, which prolonged the stage of first bloom, delayed petal fall and prevented frost damage on pistils, but led to the decrease in percent fruit set by 77%, due to ovule abortion or cessation of fruitlet development. The early and late sprays of FAME had no significant influence on either abundance of germinated pollen tubes or percent fruit set; however, the potential of late spray in improving pollen abundance and reducing pistil browning requires further investigation. In 2022, the spray of 0.01% boric acid solution led to a decrease in fruit set by 6.95%. Six-week postponement of irrigation starting from full bloom decreased soil moisture, but increased soil temperature and improved fruit set by 7.61%. The results improved our understanding about the damages of adverse spring air temperatures on pistils and ovules, and suggested the potential of irrigation adjustment in regulating soil moisture and temperature and improving fruit set in the cool and moist spring.

## 1. Introduction

As a high-value fruit product, the sustainable production of sweet cherry (*Prunus avium* L.) is significant to the global market. Successful fruit set in the early growing season is a determinant for the yield of sweet cherry at harvest. Fruit set relies on the completion of a sequence of developmental processes in the spring, including bud break, anthesis (onset of flowering, pollination, fertilization) and initial fruitlet development [1,2,3,4,5,6]. Anthesis is a developmental stage that is particularly vulnerable to temperature stresses [7]. Low air temperature may damage floral organs during their differentiation and weaken pollen viability [3,4,8,9], thereby shortening the effective pollination period and reducing the probability of successful pollination and fertilization [6]. Adverse temperatures may also cause a decline in the population of pollinators [10], and disrupt their behaviors [11] and their interactions with the plants [12,13]. Low soil temperatures and waterlogging can inhibit root activity and affect the uptake and transport of water and mineral nutrients that are essential to floral organ viability and fruit set [14,15,16]. These edaphic issues are often a consequence of heavy rainfall events in the spring, which also cause physical injury of floral organs, and lead to the loss of pollen viability and dissemination capacity due to high humidity [17]. The disrupted developmental processes consequently trigger floral drop and poor fruit set [18].

Despite the difficulty of eliminating temperature stresses, some horticultural practices can be implemented to mitigate the fruit set issue. Pollinizers [19] and pollinators [20] can substantially increase pollen availability and improve the fruit set for the cultivars with pollen-related issues. In addition, a few plant growth regulators have been reported to be effective in tree fruits. For example, aminoethoxyvinylglycine (AVG, ReTain), an ethylene inhibitor, can reduce ethylene content in floral organs and extend ovule longevity [21]. Fatty acid methyl esters (FAME) can regulate the timing of bud break; therefore, they are a potential measure to schedule anthesis to avoid adverse temperatures [22]. Pollen viability can be improved by nutrient supplements that contain boron and calcium [23,24]. Dispersions of sprayable insulating coatings of cellulose nanocrystals and nanofibrils can protect floral organs from adverse temperatures during green tip–petal fall [25]. Furthermore, orchard moisture and temperature, and vegetative and reproductive growth of *Prunus* crops, can be influenced by irrigation [26,27,28,29,30,31] and protective structure [32,33]; the effects of these horticultural measures are profound and usually dependent on cultivars and orchard environmental conditions. 

Under the changing climate, early spring weather conditions are becoming more unpredictable. Cherry flowering phenological rhythm and floral organ viability are affected by more frequent events of erratic and extreme temperatures. In the northern climate and at higher elevation where low temperature during anthesis is a predominant and frequent threat to sweet cherry fruit set, research is required to elucidate how floral organs are affected and which horticultural mitigations can effectively improve fruit set. In this study, floral buds and pistils were examined in a ‘Lapins’ sweet cherry/Krymsk 5 rootstock trial located in a frost pocket in Summerland in the Okanagan Valley, British Columbia, Canada (Site 1 at Summerland Research and Development Centre, “SuRDC1”), in comparison to four locations in the Valley with lower frost risk (“SuRDC2”, “SuRDC3”, “Oliver” and “Kelowna”; detailed description in Table 1 and Section 4.1), in the spring of 2019 and 2022. A series of experiments were conducted to investigate floral organ viability and percent fruit set under the effects of four spring horticultural mitigation measures, i.e., installation of polyethylene sleeves, FAME spray (WAIKEN, SST Australia), boric acid spray and postponed irrigation. This study aims to highlight the importance of site selection for spring frost avoidance, and to point out the necessity to evaluate the limitations of horticultural measures under specific growing conditions.

## 2. Results

### 2.1. Cold Injury on Floral Organs at SuRDC1

Phenological order of the five sites was consistent in 2019 and 2022: the earliest site was Oliver, followed by Kelowna, SuRDC3 and SuRDC2; located in a frost pocket at the foot of a hill, cooler weather delayed the anthesis at SuRDC1 (Table 1). Compared to 2019, first bloom was a week early, but anthesis was prolonged and petal fall was about 5 days late in 2022, due to a warmer March and a cooler April in the latter year (Table 2). 

SuRDC1 was the only site where night temperature dropped below −4 °C and frost risk was high after first bloom in both years. In early March of 2019, daily minimum temperature dropped to −15 °C, leading to the necrosis of floral bud primordia in about 10% of the examined 50 floral buds. Night temperature lower than −4 °C in late April of both years resulted in severe frost injury. Ratio of pistil browning in the examined flowers was 14.3% ± 4.5% on 1 May 2019 (n = 18, 30 flowers per tree), and 37.5% ± 1.8% on 6 May 2022 (n = 54, 30 flowers per tree). The percent fruit set in June was 40.9% in 2019 and 23% in 2022, significantly lower than the other four sites where no frost injury was observed on pistils in either year (Table 1; Site effect in 2019: F(4,178) = 10.22, *p* < 0.001; Site effect in 2022: F(4,169) = 35.18, *p* < 0.001). Percent fruit set in 2022 was significantly lower than in 2019 at SuRDC1, SuRDC2 and Oliver (Table 1; *p* < 0.05).

### 2.2. Effects of Elevated Temperature inside Polyethylene Sleeves

In the spring of 2019, polyethylene sleeves were wired around the bearing branches to increase ambient temperature around the floral buds (Figure 1A). The surface temperature of the floral buds surrounded by sleeves (Figure 1A,B) was 3.23 ± 0.16 °C higher than that of the untreated adjacent floral buds [Differential T_surface_ mean ± SE, n = 18, F(1,35) = 146.5, *p* < 0.001; data not graphed]. Under this setting, the stage of first bloom was prolonged and the stage of petal fall was delayed. It prevented frost damage on pistil, shown as no pistil browning in the flowers surrounded by the sleeves. However, the percent fruit set was 14.1% ± 5.4%, decreasing to about 1/3 of that on untreated branches [n = 18, F(1,35) = 20.28, *p* < 0.001; data not graphed], due to ovule abortion or cessation of fruitlet development (Figure 1C).

### 2.3. Effects of FAME Spray

The early spray (“Early”) and the late spray (“Late”) of FAME were applied about 40 days and 20 days before the anticipated normal bud break, in order to advance or set back the bud break, respectively. Early spray did not change the timing of flowering compared to Control (no spray). Late spray postponed first bloom and sepal fall for two days in three out of six experimental plots. After sample preparation and Aniline Blue staining, pollens and pollen tubes were visualized in vivo successfully (Figure 2 photos). Under Control, abundance of pollen grains on the surface of stigma was significantly higher on 6 May than on 25 April (*p* = 0.01). Compared to Control, the median of pollen abundance was higher under early spray on 25 April, and was higher under late spray on 6 May; however, the mean was not significantly different (Figure 2A, n = 6; *p* = 0.35 on 25 April, *p* = 0.61 on 6 May). Late spray led to higher pollen abundance than early spray (*p* = 0.03). The treatments did not cause difference in the number of germinated pollen tubes within the style sections on the stigma side [Figure 2B, n = 6; F(4,22) = 0.31, *p* = 0.87, n = 6], or in the presence of pollen tubes on the ovary side [F(4,26) = 1.64, *p* = 0.19; data not graphed]. Pollen tubes were present in the ovary side in 1/3 of the styles sampled on 25 April and in 5/6 of those sampled on 6 May, showing a significant timing effect [F(1,23) = 7.62, *p* = 0.01; data not graphed].

The mean of pistil browning ratio was 9.5%, 14.3% and 20.6% under late spray, control and early spray treatments (Figure 3A). The mean of percent fruit set was 36.8%, 38.2% and 40.9% under early spray, late spray and control treatments (Figure 3B). Although the differences were not significant at *p* ≤ 0.05 in either pistil browning ratio [F(2,52) = 2.10, *p* = 0.13] or percent fruit set [F(2,52) = 0.80, *p* = 0.46] (n = 18), the sprays shifted data distribution as shown in Figure 3.

### 2.4. Boric Acid Spray

The 0.01% boric acid spray led to the decrease in fruit set at SuRDC1 in the spring of 2022. Percent fruit set on the branches sprayed with 0.01% boric acid solution was 15.7% ± 0.3%, significantly lower than that of the untreated branches at 22.7% ± 0.3% [F(1,70) = 4.46, *p* = 0.04; n = 36; data not graphed].

### 2.5. Effects of Reduction in Spring Irrigation

Six-week postponement of irrigation starting from full bloom led to the decrease in soil VWC (Differential mean ± SD = 0.033 m^3^/m^3^ ± 0.012 m^3^/m^3^; range 0.006–0.067 m^3^/m^3^) (Figure 4A), the increase in soil temperature (Differential mean ± SD = 1.202 °C ± 0.523 °C; range 0.033–2.534 °C) (Figure 4B), a lower pistil browning ratio (mean ± SD 29.8% ± 1.8% versus 37.5% ± 1.8%) [Figure 4C; F(1,35) = 5.28, *p* = 0.025], and a higher percent fruit set (mean ± SD 22.4% ± 3.2% versus 14.8% ± 2.1%) [Figure 4D; F(1,35) = 4.17, *p* = 0.049].

## 3. Discussion

### 3.1. Vulnerability of Sweet Cherry Floral Organs under Variable Spring Conditions

In this study, fruit set in ‘Lapins’ sweet cherry at SuRDC1 was negatively affected by frost, precipitation and elevated ambient temperature during anthesis. Among the five sites, percent fruit set remained the lowest at SuRDC1 in both years, which was associated with pistil browning after daily minimum temperature (T_min_) during anthesis dropped below −4 °C. This suggested the importance of orchard location selection to avoid pistil injury due to spring frost. Compared to 2019, heavy precipitation and low air temperature in April 2022 at SuRDC1 led to a decrease in percent fruit set by nearly ½ (Table 1). Rainfall events could disrupt pollinator activities [13], hinder anther dehiscence and pollen release [17], and affect stigma receptivity [3], causing the failure in fertilization. Concurring with rain, low temperature in April (Table 2) could hinder pollen tube elongation and initial ovary cell division and expansion, leading to ovule abortion and fruitlet drop.

On the other hand, an increase in floral bud T_surface_ by 3.2 °C on average led to fruitlet development cessation (Figure 1C) and significantly lower fruit set on the branches surrounded by polyethylene sleeves, showing the negative consequence of simulated warming effect. Similar result was reported in Vignola and Sunburst sweet cherry, where an increase in maximum temperature by 5–7 °C resulted in an increase in average temperature by 1–3 °C and a drastic reduction in fruit set [6]. This could be attributed to rapid ovule senescence and reduced pistil viability in warm conditions [34]. This suggests that it is critical to characterize the ambient temperature dynamics and radiation conditions inside the surrounding protective mini-structures and facilities, and to avoid undesirable effects such as warming and reduced radiation when these structures are implemented in the spring. It also points out the potential impacts of global warming on cherry productivity and the necessity of implementing effective horticultural measures based on precise weather forecast to enhance fruit set resilience. In addition, it is important to define the cultivar-specific optimum temperature range for fruit set and use it to guide the site suitability evaluation prior to new cherry orchard establishment. Furthermore, compared to Mazzard rootstock, a lower fruit set was observed on Krymsk 5 rootstock (Table 1); such rootstock effect should be further investigated in continuous years in a multi-site trial that consists of different rootstocks at each location.

### 3.2. The Effectiveness of Horticultural Mitigations and Future Perspectives

The plant growth regulators that can change the readiness, viability and longevity of the floral organs can be used to improve pollination and fertilization and enhance the resilience of fruit set under adverse conditions. WAIKEN spray containing FAME was reported to effectively advance and delay the timing of bud break in the Tasmanian climate [22], suggesting its potential in scheduling anthesis to minimize the adverse temperature effect. In this study, however, neither the early spray nor the late spray altered the timing of bud break as expected. Furthermore, the sprays did not exert statistically significant effect on pollen abundance on the surface of stigma (Figure 2A), number of pollen tubes in the style sections (Figure 2B), pistil browning ratio (Figure 3A) or percent fruit set (Figure 3B). The ineffectiveness could be attributed to the low temperatures on 7 March 2019, the day of early spray application, which had affected the efficacy of the spray product. The daily T_min_ in early-mid March in the Okanagan region is usually close to 0 °C, which would be a persistent limitation to the use of the spray 35–50 days before bud break to advance anthesis. On the other hand, late spray showed potential in improving the pollen abundance (Figure 2A) and reducing pistil browning (Figure 3A). The statistical significance of late spray may have been underestimated due to the limited number of replications (n = 6 flowers), as statistical power depends on both effect size and sample size, and it is more likely to detect a smaller effect with a larger sample size. Within the recommended late spray threshold between 20 days before bud break and green tip (manufacturer’s guide for WAIKEN Orchard Spray Emulsion Concentrate, SST Australia), sequential sprays at a 3-day incremental interval can be conducted to identify the best spray timing that can set back the bud break most effectively in the local climate of interest, as a measure to avoid frost and improve fruit set.

Boron is a critical element for the success of flowering and fruit set. Appropriate boron level can promote pollen germination and pollen tube elongation. Inappropriate concentration, formula, and application timing and method can cause deficiency, ineffectiveness or toxicity [35,36]. In this study, 0.01% boric acid spray negatively affected fruit set, suggesting that the acidic form in such concentration might not be a beneficial practice at full bloom, although it effectively promoted pollen germination and elongation in the in vitro study [23,24]. Tissue acidification [37] and high moisture retention [17] were reported to reduce pollen viability and anther dehiscence in fruit trees. Further study is required to elucidate whether the boric acid spray affects the viability of floral organs due to these issues, to investigate the appropriate concentration and form of boron, the timing, and the co-effects of calcium and nitrogen supplements in aerial spray, and to compare its effectiveness with fertigation.

Irrigation scheduling plays a critical role in determining the dynamics of water availability and soil temperature in the rhizosphere. Its effects on cherry tree hardiness, vigor and yield have been reported in several studies [38,39,40], whereas the effect on fruit set was yet to be elucidated. In the spring, irrigation can influence the root activity in transporting water and nutrients that are essential to flowering and fruit set, such as calcium, boron and zinc [41,42,43,44,45]. It can also affect the timing of dormancy break, the development of floral organs, the early vegetative growth and the abundance of carbohydrates for reproductive growth in woody plants [14,15]. Water stress can induce phytohormone imbalance in abscisic acid, auxin, gibberellin and ethylene, which dynamics can determine the formation of abscission zone at the base of pedicels, causing flower and fruit drop [46]. In this study, postponed irrigation during anthesis in the moist and cool spring of 2022 led to an increase in soil temperature without causing water deficit (Figure 4A,B), and an increase in percent fruit set which was associated with less pistil browning (Figure 4C,D). This is consistent with the study of Greer et al. which showed that warmer soil temperatures could promote bud break and early season development in apples, possibly attributed to higher root activity and better nutrient supply in the warmer rhizosphere [47]. However, Hammond and Seeley’s earlier work showed that soil warming did not affect the spring bud development in the studied *Malus* or *Prunus* species [48]. This suggests that the root responses to warmer soil temperatures may be dependent on the interactions of other environmental factors that require more investigation. In addition, postponed irrigation in the wet spring may improve soil aeration in rhizosphere [16,49]. Research attention should also be drawn to the relation of water content and frost susceptibility of floral organs [50,51] under postponed irrigation. Irrigation scheduling shows great potential in improving fruit set, particularly in the early- and late-flowering cultivars grown at the geographic extremes which are more susceptible to the risks of adverse temperatures. However, it should be carefully tailored to the specific cultivars, the edaphic conditions of the orchard and the spring weather conditions that vary each year. Future studies should elucidate the soil water potential range during anthesis to achieve optimal fruit set.

## 4. Materials and Methods

### 4.1. Site Description and Fruit Set Monitoring

For treatment implementation, the main experimental site of ‘Lapins’ sweet cherry on ‘Krymsk 5′ rootstock was located in the frost pocket at the elevation of 487 m, in the experimental farm of Summerland Research and Development Centre, Agriculture and Agri-Food Canada, Summerland, the Okanagan Valley, British Columbia, Canada (Site “SuRDC1”, 49.5657° N, 119.6365° W). The trial of 216 trees in 6 rows was established in 2015 in loamy sandy soil. Tree canopy was pruned to Tall Spindle Axes structure [52] and kept at the height of 2.2–2.5 m. Water was supplied through micro-sprinklers. Granular fertilizer (TerraLite 22-5-12) was casted under the canopy in June at the rate of 305 g per tree. Each plot (experimental unit) consisted of 3 sampling trees in the centre of the plot, with 1 buffer tree on each side of the plot; plots were distributed by completely randomized block design in the trial; treatment details of each horticultural mitigation trial were described in Section 4.2. 

For phenology and fruit set monitoring, 18 trees were distributed in 6 control plots at SuRDC1 (3 trees/plot), in comparison to the other four monitored sites of lower frost risk, with 2 sites at lower elevation in the experimental farm (“SuRDC2”, 49.5642° N, 119.6396° W, 16 trees; “SuRDC3”, 49.5653° N, 119.6418° W, 9 trees), 1 in Oliver (“Oliver”, 49.17° N, 119.56° W) and 1 in Kelowna (“Kelowna”, 49.88° N, 119.37° W) (10 trees each site; precise geo-coordinates were omitted for growers’ sites) (Table 1). Air temperature was monitored using HOBO sensors to detect daily T_min_ (MX2301A, Onset/HOBO). T_sum_ and precipitation data were acquired from farmwest.com (Summerland EC station for SuRDC sites, Olive Central for Oliver site, Kelowna East for Kelowna site; Table 1 and Table 2).

Phenology of floral developmental stages from green tip (BBCH54) to sepal fall (Green Ovary, BBCH72) [53] was recorded on the tagged branches in each sampling tree (1 branch per tree in 2019 and 2 branches per tree in 2022 at SuRDC1, 1 trunk section per tree at SuRDC2, 3–4 branches per tree at SuRDC3, and 3 branches per tree at Kelowna and Olive; Table 1). At full bloom, flower samples were collected from adjacent untagged branches to examine pistil browning (5 flowers per tree). Flower counts at white tip—full bloom and fruitlet counts after shuck fall were recorded on the tagged branches; percent fruit set was estimated as fruitlet counts divided by flower counts for each branch and analyzed for statistical significance of site effect by ANOVA (n = number of branches per site, Tukey’s pairwise comparison, *p* ≤ 0.05). In addition, at SuRDC1, 50 random floral buds were sampled on 7 March 2019 and dissected to examine primordia, following the cold snap on 3–4 March (daily T_min_ below −15 °C).

### 4.2. Spring Horticultural Mitigations at SuRDC1

#### 4.2.1. Installation of Polyethylene Sleeves to Increase Floral Bud T_surface_

In the spring of 2019, 36 polyethylene sleeves (made of Uline 6 mil heavy duty poly tubing, about 0.152 mm thick) wrapped onto plastic racks were wired around the bearing branches to surround the floral buds, from bud break to sepal fall (2 sleeves on each tree, 18 trees; Figure 1A). The two ends of the polyethylene sleeves were kept open to allow air flow and pollinator activities. Two tagged branches on the same trees were monitored as the control. Thermal images of the floral buds were captured at 12:00–12:30 PM of 17 April, using FLIR E8 Infrared camera (FLIR^®^ Systems Inc., Wilsonville, OR, USA; 7 mm focal lens, 320 × 240 IR resolution); On the surrounded branches, the photos were taken through the open end of the sleeve (Figure 1B). Infrared and RGB channels were separated in software FLIR Tools V. 6.4 (FLIR^®^ Systems Inc.); the RGB image (Figure 1B bottom) was referred to locate the floral buds in the corresponding infrared image (Figure 1B top). T_surface_ of floral buds was automatically computed in FLIR Tools; T_surface_ of 3 buds was averaged to represent the mean bud T_surface_ for each sleeved and tagged branch. Flowering stages, pistil browning and aborted ovule (Figure 1C) were recorded for each branch. For T_surface_ and percent fruit set, the mean of the two branches per treatment per tree was calculated and analyzed for statistical significance of polyethylene sleeve effect by ANOVA (n = 18 trees, Tukey’s test, *p* ≤ 0.05).

#### 4.2.2. FAME Spray to Regulate Bud Break

The efficacy of an orchard spray emulsion concentrate which contains FAME (WAIKEN, SST Australia; active constituent: 388 g/L FAME including 10–35% dibutyl phthalate, 10–20% ethoxylated nonylphenol, 1–12% oxirane methyl polymers), was tested as a run-off spray of 1:24 dilution (1 L of product diluted in 24 L water to make 25 L working spray; about 15.5 g/L of FAME in the working spray) in the spring of 2019 at two time points, to adjust the timing of bud break. The early spray (“Early”) was applied on 7 March, about 40 days before the anticipated normal bud break, to advance bud break; T_min_ and T_max_ (maximum temperature) were −4.3 °C and 2.3 °C on this day. The late spray (“Late”) was applied on 1 April, about 20 days before normal bud break, to set back the bud break; T_min_ and T_max_ were 1.3 °C and 15.8 °C on this day. No spray was applied to the control trees (“Control”). These treatments aimed to expose the flowers to different temperatures, and to investigate its impacts on pollen viability and percent fruit set. Each treatment was applied to 18 trees, with 3 trees per plot, 6 plots randomized in 6 rows (1 plot per row). On each tree, 2 branches were tagged for phenological observation and their mean percent fruit set was calculated; 7 flowers per tree were examined for pistil browning. Statistical significance of FAME spray effects was analyzed by ANOVA (n = 18 trees, Tukey’s pairwise comparison, *p* ≤ 0.05).

To examine the abundance of pollen grains landing on the stigma and the number of germinated pollen tubes in the style, six flowers were sampled for early spray on 25 April, for Control on 25 April, 1 May and 6 May each, and for late spray on 6 May respectively (n = 6 flowers). Pistils were preserved in Formaldehyde Alcohol Acetic Acid fixative (FAA; 50% ethanol, 5% (*v*/*v*) acetic acid, 3.7% (*v*/*v*) formaldehyde) at 4 °C until analysis. For sample preparation, stigma was cut apart from style using a sharp razor blade and set aside to analyze the pollen grains on its surface. The style was cut apart from the ovary, placed on a piece of double-sided tape to hold it down, and then pressed gently with a clean glass slide to flatten it slightly in order to make a straight cut; both ends of the style were cut off transversely about 0.5 mm from the ends and then dissected longitudinally for better dye penetration to stain the pollen tubes within the style. Both the stigma and the style were placed in Aniline Blue solution (0.1% Aniline Blue *w*/*v* in 0.1 mol L^−1^ KH_2_PO_4_, Sigma-Aldrich) for 30 min (modified based on Lu [54]). Using a transfer pipet, the stigma was positioned in the centre of a depression slide with the top pointing upwards. The four sections from each style (2 longitudinal sections from the stigma end and 2 from the ovule end) were placed on a coverslip facedown; the coverslip was then placed on the slide with the cut sections facing up. Sample images were acquired using Zeiss Axio Imager M2 wide-field microscope (Zeiss). Stigma images were acquired under 5× objective lens, with the microscope light source blocked out; an external warm white light was used to illuminate the sample from the side to provide better light diffusion and contrast. The images of style sections were acquired under 10× objective lens in green fluorescence channel under UV excitation. For both the stigma and the style images, multiple images were taken for each sample and stacked into one image with clear focus using Helicon Focus software (HeliconSoft.com). The counting tool in Image J [55] was used to count the number of pollen grains on the surface of stigma (Figure 2A), and the number of pollen tubes from both style sections in the stigma end which were summed up to estimate the number of germinated pollens (Figure 2B). In the ovary end of the style, the presence of pollen tubes was recorded. Statistical significance of FAME spray effects was analyzed by ANOVA (n = 6 flowers, Tukey’s test, *p* ≤ 0.05).

#### 4.2.3. Boric Acid Spray

The 0.01% boric acid solution (0.099 g H_3_BO_3_ in 1 L of distilled water, Sigma-Aldrich) was sprayed to 1 branch per tree of 36 trees on 3 May 2022 at full bloom (with 3 trees per plot, 12 plots randomized in 6 rows, 2 plots per row). Flower counts at white tip–full bloom and fruitlet counts after shuck fall were recorded on treated branches and adjacent control branches of the same trees; percent fruit set was estimated as described in Section 4.1 and analyzed for statistical significance of boric acid spray effect by ANOVA (n = 36, Tukey’s test, *p* ≤ 0.05).

#### 4.2.4. Postponed Irrigation and Monitoring of Soil Moisture and Temperature

Micro-sprinkler irrigation was scheduled for two hours every 72 h from full bloom to the end of September. When daily maximum temperature exceeded 30 °C in July and August, irrigation was scheduled for two hours every 48 h. In the moist and cool spring of 2022, irrigation was stopped between 28 April and 7 June (Postponed Irrigation) in and around 6 plots randomized in 6 rows (1 plot per row, 3 trees per plot, n = 18 trees). One control plot of 3 trees was located a plot apart from the Postponed Irrigation plot in the same row (n = 18 trees). The 5TM soil sensors (Meter Environment, Pullman, WA, USA) were installed in 2 Postponed Irrigation plots and 2 control plots, to monitor volumetric water content (VWC) and soil temperature in the topsoil of 20 cm depth at 30 min interval (EM50 data loggers, Meter Environment). Flower counts, fruitlet counts and percent fruit set were recorded on 2 tagged branches per tree as described above.

### 4.3. Data Analysis

Statistical analysis and graphing were conducted using OriginPro 8.0 (OriginLab, Northampton, MA, USA). For the fruit set monitoring, significant difference among the five sites was assessed by ANOVA (*p* ≤ 0.05, Tukey’s test for pairwise comparison, site as the fixed effect). For each horticultural mitigation trial, significant difference between control and treatment was assessed by ANOVA (*p* ≤ 0.05, Tukey’s test, treatment as the fixed effect, outliers were excluded). Mean ± standard error were shown along with F[df (degree of freedom), N (total samples minus df)] and *p* values for statistical significance of the fixed effects in comparison. In the horticultural mitigation trials, individual flowers were analyzed as biological entity for the pollen abundance on stigma and the number of germinated pollen tubes (n = the number of sampled flowers). Sampling trees were analyzed as entity for ratio of floral organ damage, floral bud T_surface_ and percent fruit set (n = the number of trees; the mean of branches under each treatment on each tree was calculated to represent the tree).

## 5. Conclusions

The fruit set in the ‘Lapins’/Krymsk 5 planting located in a frost pocket was negatively affected by frost, precipitation and elevated ambient temperature during anthesis. Low percent fruit set was associated with pistil browning when T_min_ was below −4 °C, and with fruitlet development cessation when floral bud T_surface_ was elevated by about 3 °C. Postponed irrigation during anthesis in the moist and cool spring improved fruit set, suggesting irrigation adjustment as an effective measure to mitigate fruit set issue when there is excessive moisture. Further investigation on spray contents and timing is required to improve the efficacy of FAME and boron supplements. The FAME spray showed potential in delaying anthesis and preventing frost injury; its physiological mechanism and horticultural effects under the climate of interest wait to be elucidated with more replications using the in vivo pollen and pollen tube examination method documented in this study.

## Figures and Tables

**Figure 1 plants-12-00468-f001:**
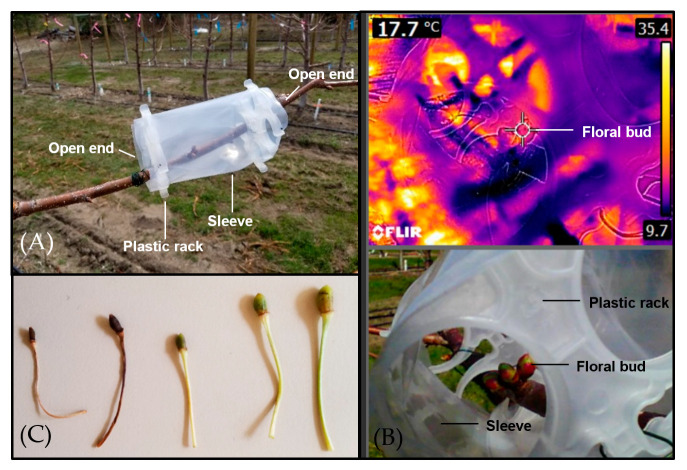
The installation of polyethylene sleeves to increase floral bud surface temperature (T_surface_) in a ‘Lapins’/Krymsk 5 trial at SuRDC1. (**A**) The set-up of polyethylene sleeves around floral buds. (**B**) Floral bud T_surface_ measurement using an infrared imager (photo was taken through the open end of the sleeve; top: infrared channel, bottom: RGB channel). (**C**) Ovule abortion and cessation of fruitlet development under the setting.

**Figure 2 plants-12-00468-f002:**
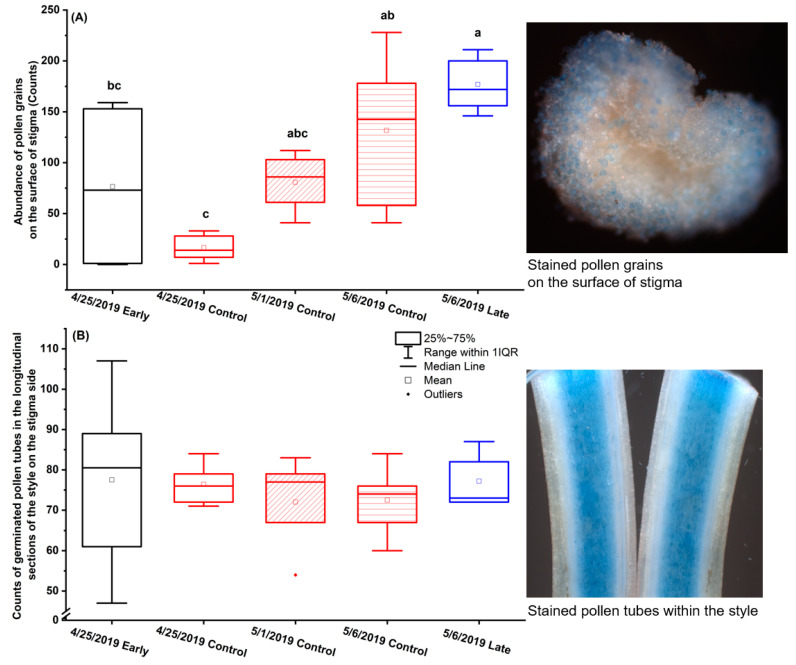
Impacts of fatty acid methyl esters (FAME) on (**A**) pollen abundance on the surface of stigma and (**B**) number of germinated pollen tubes in the longitudinal sections of the style on the stigma side of ‘Lapins’ sweet cherry in the spring of 2019 at SuRDC1. The early spray (Early) was applied on 7 March and flowers were sampled on 25 April. The late spray (Late) was applied on 1 April and flowers were sampled on 6 May. Flowers of Control treatment were sampled on 25 April, 1 May and 6 May. Pollen grains and pollen tubes were stained with Aniline Blue (n = 6 flowers). Boxplots show median (horizontal line) and interquartile ranges. In (**A**), the letters stand for significant difference at *p* < 0.05 [ANOVA, Tukey’s test for pairwise comparison, F(4,22) = 6.97, *p* < 0.001].

**Figure 3 plants-12-00468-f003:**
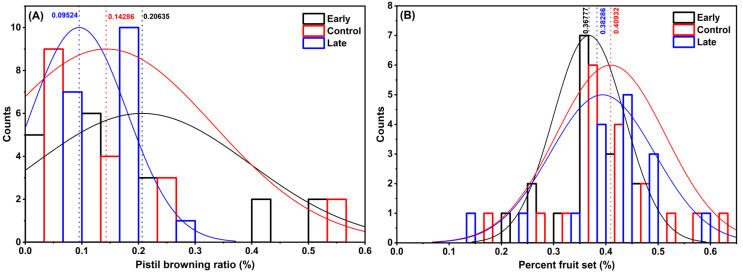
Impacts of FAME on (**A**) pistil browning ratio and (**B**) percent fruit set of ‘Lapins’ sweet cherry in the spring of 2019 at SuRDC1. The early spray (Early) and the late spray (Late) were applied on 7 March and 1 April, respectively. Solid lines in the histograms are distribution curves (n = 18 trees per treatment). Vertical dot lines stand for the mean for each data group.

**Figure 4 plants-12-00468-f004:**
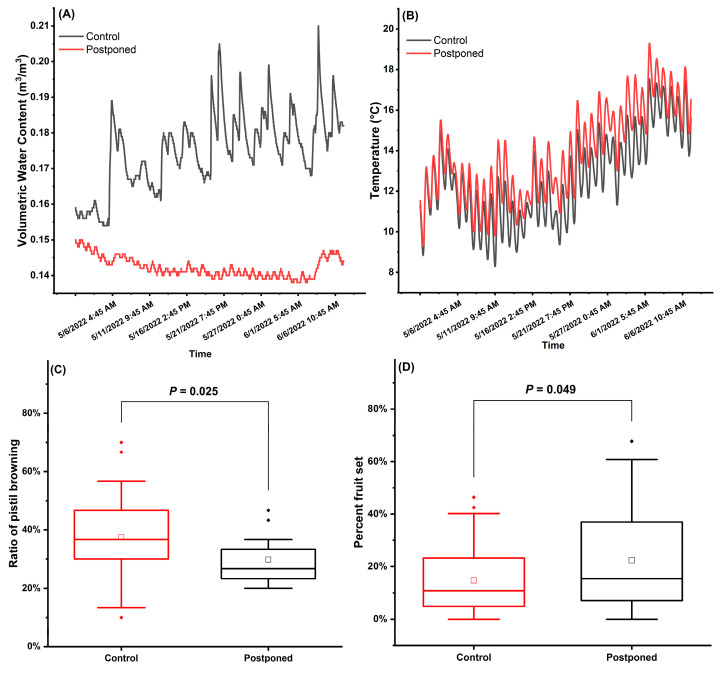
Postponed irrigation led to (**A**) the decrease in Volumetric Water Content (VWC) in topsoil, (**B**) the increase in soil temperature, (**C**) the decrease in pistil browning and (**D**) the increase in fruit set. Boxplots in (**C**) and (**D**) showed median (horizontal line), mean (square) and interquartile ranges; significant difference is shown as p value for each pair of comparison (AVONA, Tukey’s test, *p* < 0.05, n = 18 trees per treatment).

**Table 1 plants-12-00468-t001:** Site description, phenology and percent fruit set of ‘Lapins’ sweet cherry at five monitored sites in the Okanagan Valley, British Columbia, Canada, in the spring of 2019 and 2022.

Sites	Elevation, Frost Risk	Rootstock, Canopy Structure	Replications for % Fruit Set	2019				2022				Year Effect
				First Bloom–Petal Fall	T_min_ (°C)	PP (mm)	% Fruit Set	First Bloom–Petal Fall	T_min_ (°C)	PP (mm)	% Fruit Set	
SuRDC1	487 m, High risk	Krymsk 5, TSA	6 plots, 3 trees/plot, 1–2 branches/tree	27 April –7 May	−4.10	3	40.9 ± 2.5 c	21 April –12 May	−4.12	12	23.0 ± 2.5 c *	F(1,53) = 20.06 *p* < 0.001
SuRDC2	416 m, Low risk	Krymsk 5, SSA	4 plots, 4 trees/plot, 1 trunk section/tree	24 April –3 May	−0.16	3	46.0 ± 1.9 bc	17 April –9 May	−1.72	12	35.8 ± 1.8 b *	F(1,31) = 8.90 *p* = 0.004
SuRDC3	424 m, Low risk	Mazzard, Open heart	9 trees in 4 rows, 3–4 branches	23 April –3 May	−0.98	3	59.2 ± 2.2 a	15 April –8 May	−2.10	12	66.4 ± 2.9 a	F(1,59) = 3.74 *p* = 0.06
Oliver	332 m, Low risk	Mazzard, Central leader	2 plots, 5 trees/plot, 3 branch/tree	17 April –30 April	−1.07	3	55.2 ± 0.3 a	11 April –4 May	−2.27	3	36.6 ± 1.8 b *	F(1,59) = 60.51 *p* < 0.001
Kelowna	483 m, Low risk	Mazzard, Central leader	2 plots, 5 trees/plot, 3 branch/tree	19 April –2 May	−1.27	1	53.1 ± 0.2 ab	12 April –6 May	−2.46	2	57.3 ± 3.5 a	F(1,59) = 0.84 *p* = 0.36

Note: TSA and SSA stands for Tall Spindle Axes and Super Slender Axes canopy structures, respectively. Plots were randomized in each trial. T_min_ and PP stand for minimum air temperature and moisture deficit during full bloom–petal fall, respectively. Different letters in the same column of year|% fruit set represent significant difference among sites in each year; asterisks in the same row of site|%fruit set stand for significant difference between years at each site (*p* < 0.05, one-way ANOVA, Tukey’s test for pairwise comparison, n = number of branches per site).

**Table 2 plants-12-00468-t002:** The accumulated mean daily temperatures above 0 °C (T_sum_) in March and April in 2019 and 2022 in Summerland, Oliver and Kelowna (British Columbia, Canada).

Sites	Weather Station Location	T_sum_ in March (°C)			T_sum_ in April (°C)		
		Historical Average	2019	2022	Historical Average	2019	2022
Summerland EC	Latitude: 49.5700°; Longitude: −119.6769°; Elevation: 454 m	143.62	108.2	192.4	269.9	275.5	215.8
Oliver Central	Latitude: 49.1575°; Longitude: −119.5514°; Elevation: 349 m	163.9	150.02	214.87	280.88	317.84	217.37
Kelowna East	Latitude: 49.8738°; Longitude: −119.4436°; Elevation: 432 m	148.83	144.93	182.84	268.54	278.88	213.7

Note: Data were acquired from the weather stations in adjacency to the cherry sites (Farmwest.com; accessed on 12 December 2022). The average of T_sum_ was the historical average of 61 years at Summerland EC station, and of 52 years at Oliver Central station and Kelowna East station.

## Data Availability

Data are available upon request.

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
