# Peer review of "Horticultural Practices in Early Spring to Mitigate the Adverse Effect of Low Temperature on Fruit Set in ‘Lapins’ Sweet Cherry"

_plants, 2023, doi:10.3390/plants12030468_

Round 1

Reviewer 1 Report

This is a carefully designed and nicely implemented study aiming at choosing the best frost protection for the floral buds and developing flowers and fruits in sweet cherry. The only concern is that about the presentation of some results (please see specific notes below). Some information about the methods is also missing and should be provided.

L13: …to investigated… => …to investigate…

L15: which fatty acid(s)? Please specify the preparation origin, also in the methods.

It would be desirable to have a table with the list of experimental sites, number of trees etc. also in the Methods to have a clear idea of the experimental design. Alternative: amend Table 1 with n of trees.

Please specify the actual concentration of the FAME, not just dilutions + see related comment above.

Fig. 1B: it is not clear when the thermal picture was captured. It is also difficult to understand what is there and where are the buds etc. Annotating the figure might help.

Fig. 3: what do the smooth curves represent? Distribution shape?

Author Response

Dear Reviewer,

Thank you very much for reviewing our manuscript plants-2131839 and for providing valuable revision suggestions. We carefully revised the manuscript accordingly and prepared the following responses to your comments. The changes have been tracked in the revised manuscript. Please note that the line numbers have changed, and the line numbers listed below correspond to the Word file in the All Markup mode.

Thank you again for your generous help with improving this manuscript.

Yours sincerely,

Hao on behalf of the coauthors Ms. Danielle Ediger and Mehdi Sharifi

Reviewer’s comments and suggestions for authors

This is a carefully designed and nicely implemented study aiming at choosing the best frost protection for the floral buds and developing flowers and fruits in sweet cherry. The only concern is that about the presentation of some results (please see specific notes below). Some information about the methods is also missing and should be provided.

1. L13: …to investigated… => …to investigate…

Response: We corrected it in Line 16.

2. L15: which fatty acid(s)? Please specify the preparation origin, also in the methods.

Response: It is WAIKEN (SST Australia). We clarified in Line 20, and added the full details in Line 397-401.

3. It would be desirable to have a table with the list of experimental sites, number of trees etc. also in the Methods to have a clear idea of the experimental design. Alternative: amend Table 1 with nof trees.

Response: We revised the site description in Line 342-371. We clarified that the treatments were implemented at site SuRDC1 only, in completely randomized block design. Fruit set observation and phenological monitoring were conducted at five sites. The number of replications was added in the text and in Table 1.

4. Please specify the actual concentration of the FAME, not just dilutions + see related comment above.

Response: The details were provided in Line 397-401.

5. 1B: it is not clear when the thermal picture was captured. It is also difficult to understand what is there and where are the buds etc. Annotating the figure might help.

Response: We added more details about how thermal images were captured in Line 382-387, and added annotations in Figure 1A.

6. F3: what do the smooth curves represent? Distribution shape?

Response: Yes, they are distribution curves of the data sets. We clarified in the figure caption in Lin 195-196.

Reviewer 2 Report

Dear Authors,

This manuscript provides new insights into changes in floral and fruit biology in sweet cherries under different horticultural practices employed to mitigate plant productive problems as affected by low-temperature stress. However, the manuscript needs major and minor revisions before publication in Plants Journal, as follows:

Major revisions

Please review the statistical procedure used in the M&M and Result section, including more explanation of the experimental design (complete random or blocks, replicates) used in the trials (line 317, line 335, line 355, line 383, line 391). Also is quite important to review the statistical analysis employed to show the results. Why did not apply ANOVA analysis to all trials? The lack of clarity of the experimental designs and statistical procedures used in each trial does not allow the results of this research to be fully conclusive.

Minor revisions

Line 261, please delete the trademark of the product “Waiken®”

Line 273, please explain more clearly how the limited replicates affect the results.

Line 275, please delete the trademark of the product “Waiken®”

Line 286, please include a citation to support the argument

Line 301 – 305, please include a citation to support the argument

Author Response

Dear Reviewer,

Thank you very much for reviewing our manuscript plants-2131839 and for providing valuable revision suggestions. We carefully revised the manuscript accordingly and prepared the following responses to your comments. The changes have been tracked in the revised manuscript. Please note that the line numbers have changed, and the line numbers listed below correspond to the Word file in the All Markup mode.

Thank you again for your generous help with improving this manuscript.

Yours sincerely,

Hao on behalf of the coauthors Ms. Danielle Ediger and Mehdi Sharifi

Reviewer’s comments and suggestions for authors

Dear Authors,

This manuscript provides new insights into changes in floral and fruit biology in sweet cherries under different horticultural practices employed to mitigate plant productive problems as affected by low-temperature stress. However, the manuscript needs major and minor revisions before publication in Plants Journal, as follows:

Major revisions

1. Please review the statistical procedure used in the M&M and Result section, including more explanation of the experimental design (complete random or blocks, replicates) used in the trials (line 317, line 335, line 355, line 383, line 391). Also is quite important to review the statistical analysis employed to show the results. Why did not apply ANOVA analysis to all trials? The lack of clarity of the experimental designs and statistical procedures used in each trial does not allow the results of this research to be fully conclusive.

Response: We apologize for the ambiguity. We revised 4.3 Data Analysis Line 463-471 to clarify that ANOVA was applied to all trials (Tukey’s test). In the figures, we annotated where the difference was significant at P < 0.05. In the text, we described the F value [df (degree of freedom), N (total samples minus df)] and P value for statistical significance of the fixed effects in comparison by ANOVA.

We also revised the site description in Line 342-371. We clarified that fruit set observation and phenological monitoring were conducted at five sites; the treatments were implemented at site SuRDC1 only, in completely randomized block design. We added the number of replications in the text of Results and in Table 1.

Minor revisions

2. Line 261, please delete the trademark of the product “Waiken®”

Response: Deleted.

3. Line 273, please explain more clearly how the limited replicates affect the results.

Response: We added some explanation in Line 290-293 that “the statistical significance of late spray effect significance may have been underestimated by due to the limited number of replications (n = 6 flowers); as statistical power depends on both effect size and sample size, and it is more likely to detect a smaller effect with a larger sample size”.

4. Line 275, please delete the trademark of the product “Waiken®”

Response: Deleted.

5. Line 286, please include a citation to support the argument

Response: The argument was revised as a speculation. We added two citations (17, 37) to support, in Line 303-306.

6. Line 301 – 305, please include a citation to support the argument

Response: We revised the argument, added more discussion and 6 citations (16, 47-51) to support the discussion, in Line 322-332.

Round 2

Reviewer 2 Report

Dear Authors, 

The new version of manuscript was carefully revised. The incorporation  and clarification of statistical analysis satisfactorily improved the manuscript. 

Kind regards 

Reviewer 2